# Child-to-Parent Violence, Peer Victimization and Cybervictimization in Spanish Adolescents

**DOI:** 10.3390/ijerph18179360

**Published:** 2021-09-04

**Authors:** Paula López-Martínez, David Montero-Montero, David Moreno-Ruiz, Belén Martínez-Ferrer

**Affiliations:** 1Department of Education and Social Psychology, Pablo de Olavide University, 41013 Seville, Spain; dmonmon@alu.upo.es (D.M.-M.); bmarfer2@upo.es (B.M.-F.); 2Department of Social Psychology, Valencia University, 46003 Valencia, Spain; david.moreno-ruiz@uv.es

**Keywords:** child-to-parent violence, peer victimization, cybervictimization, adolescence

## Abstract

The aim of this study was to analyse the relationship between child-to-parent violence (CPV) (high, moderate and low), peer victimization (PV) (relational and overt, both physical and verbal) and cybervictimization (CV) (relational and overt), taking into account the role of sex. 1304 adolescents (53.14% girls) between the ages of 11 and 18 enrolled at secondary schools in the Autonomous Communities of Valencia, Aragón and Andalusia participated in the study. Adolescents with high CPV scores obtained higher scores for all types of PV and CV compared to the other CPV groups. Boys scored higher than girls in overt physical PV and in overt CV and girls obtained higher scores in relational PV. A statistically significant interaction effect was observed; boys with high CPV scores reported greater overt CV. The results suggest the importance of CPV in relation to specific forms of PV and CV and highlight the need to take into account the different processes of family socialization between boys and girls to reduce the likelihood of adolescents being victimized.

## 1. Introduction

In recent years, violent behaviour by adolescents, directed against both parents and peers, has increased considerably [1,2]. Some studies have reported commonalities between child-to-parent violence (CPV) and peer violence [3,4]. Several studies also support the link between peer victimization (PV) and subsequent delinquency [5,6]. However, there are still few studies that analyse CPV in relation to suffering from cybervictimization (CV) and PV. This research aimed to study the relationship between PV, CV and CPV based on sex.

### 1.1. Child-to-Parent Violence

CPV is defined as the harmful acts committed by adolescents against either of their parents or others who perform this function in order to control, dominate and have power over them [7]. Violence can be physical, psycho-emotional and/or economic and is carried out repeatedly and over time [8,9]. It usually begins with economic violence before progressing to psychological and even physical violence, so that all three forms of violence end up appearing simultaneously [10]. This behaviour represents a growing social problem, since the number of parents reporting such behaviour in their children has increased by 101.1% in recent years [11]. Furthermore, many cases are not reported; some parents underestimate violence by their children and feel uncomfortable admitting this situation, since it is common for society to interpret CPV as a failure to educate and set limits [12,13].

### 1.2. Victimization

PV is conceptualized as the negative experience of being the object of overt (physical and verbal) and relational aggression by other peers that takes place in and around the school, particularly in places with little adult supervision [14,15].

Specifically, overt PV implies suffering direct violence [16], either through verbal aggression, when the victim is insulted or threatened [17] or physical aggression, such as being beaten or pushed [18]. Relational PV refers to suffering a more subtle type of violence, aimed at causing harm in one’s circle of friends or undermining one’s sense of inclusion in the group by spreading rumours or withdrawing friendship [19]. The victim is perceived to be at a disadvantage with respect to the aggressor and finds it very difficult to get out of the situation [20,21].

In response to the attacks suffered, most victims react passively, displaying withdrawal and submission, while a minority react aggressively, seeking revenge [22,23]. Passive victims of school violence tend to have low self-esteem, depressive symptoms, high levels of stress and dissatisfaction with life and, occasionally, especially among victims who exercise the role of aggressor, low levels of self-control. This low self-control implies that they do not think about the consequences of their behaviour. They also participate more in behaviours that irritate and provoke their peers and receive less social support, which means that they do not have friends who protect them from bullying [24]. In this sense, victimised people perceive little support, are more vulnerable and are rejected by their peers [25,26]. Another observed aspect is that victims do not know how to compensate for emotional reactions occurring in difficult social interactions. In fact, adolescents with more experience in aggression, because they are either bullies or victims, are more likely to attribute hostile intentions to the behaviours of others, even if they are ambiguous [27].

Regarding socio-demographic factors such as sex, there is no consensus in the case of PV. Many authors agree that it occurs more frequently in boys [22,28,29]. However, other studies have reported that girls suffer PV to a greater extent than boys [30,31].

### 1.3. Cybervictimization

In recent years, the incorporation of information and communication technologies (ICTs) in everyday life has given rise to a virtual culture in which adolescents participate and to which they contribute. These technological advances have enabled the development of innovative communication channels and forms, but they have also been used to generate new forms of victimization. CV occurs when the victim is harassed through electronic or digital devices and in virtual environments through hostile or aggressive messages intended to cause the victim harm or discomfort [32,33]. CV transcends the purely physical sphere as it is not restricted exclusively to the school context, surpassing that barrier to continue in any place where the adolescent has access to an electronic device connected to the Internet [34,35].

PV and CV are closely related and have common characteristics; in both cases the victims suffer intentional and persistent harassment by the aggressor [36,37]. Victims of cyber-bullying are often also victims of traditional bullying [38,39]. However, in CV, victimization can occur at any time, since the messages are received on an electronic device, so there are no safe places for the victim where they can flee or hide [6]. Moreover, the scope and breadth of the audience is potentially much greater than in the case of PV and the attacks can be reproduced indefinitely since they are stored in cyberspace [37,40]. The aggressor’s identity remains hidden to the victim of CV, thus favouring disinhibition when committing the aggressions, coupled with moral disconnection and a lack of empathy and remorse regarding the victim’s situation [41,42].

In terms of the sex factor in CV, most studies report that girls are more cybervictimized [35,38,39]. Additionally, they are more involved in the problematic use of virtual social networks and participate more in activities that involve social interaction, such as chats, which involve greater exposure to conflictive situations [28].

### 1.4. Child-to-Parent Violence, Peer Victimization and Cybervictimization

The family is one of the main socializing agents for adolescents. Many studies have examined the family characteristics of victims of PV; a positive family climate, with affection and communication, is related to the development of personal skills by adolescents, which makes them less vulnerable to victimization and has a buffer effect on the distress suffered by victims [26,43,44,45]. However, the perception of family conflict contributes to the fact that adolescents are victims of victimization, since they tend to perform submissive behaviours and present themselves to their peers as vulnerable and easy targets for abuse [46]. Victims of PV define their parents as cold, indifferent, hostile, in some cases overprotective or permissive and feel rejected and little supported by them [47,48]. In addition, communication problems and family expressiveness have been observed in victims, increasing the psychological discomfort of adolescents, who find it more difficult to deal effectively with aggression [12,32]. This process is bi-directional; greater PV also implies greater psychological distress, which in turn makes adolescents have worse communication with their parents [20]. However, positive parent-adolescent relations, including positive communication and disclosure between parents and children implies better psychosocial adjustment and a lower level of victimization among peers [33,47].

Victims of CV report negative family functioning, with family conflicts, poor communication between parents and children, little family cohesion and lack of emotional support from parents [38,39,49].

Adolescents exercising CPV and those suffering from PV both have common family problems, such as a family climate characterized by conflict and low cohesion, dysfunctional family dynamics, lack of communication and affective and emotional deficiencies [50,51]. Some studies indicate that the perception of family conflict implies in adolescents submissive and internalizing reactions to a greater extent than aggressive or externalizing behaviours [44,45,52,53]. However, other studies conclude that adolescents who exercise CPV are often victims of PV and that this exposure to violence at school fuels aggression towards parents [7,54,55]. Loinaz et al. [56] reported that this relationship was more frequent in the case of girls; female aggressors engaging in CPV were victims of PV to a greater extent than boys who exercised CPV.

Likewise, few studies have studied the link between CPV and CV. CPV is related to the problematic use of virtual social networks and difficulties in the use of the Internet, especially in the case of girls, who seek new socialization environments where personal relationships are more satisfactory, especially when there are deficits in other socialization settings, such as family [57]. However, research in this field is in an incipient state, especially with regard to the role played by CPV in relation to specific types of PV and CV, i.e., relational, overt, physical and verbal.

### 1.5. The Present Study

The aim of this comparative cross-sectional study was to analyse the relationship between CPV and specific dimensions of PV and CV, also taking into account the role of sex.

The following hypotheses were proposed:

**Hypothesis** **1** **(H1).** 
*Adolescents with high levels of CPV will suffer more PV, both relational and overt, physical and verbal, as well as more CV, both relational and overt.*


**Hypothesis** **2** **(H2).** 
*An interaction effect will be observed between CPV and sex, whereby the group of boys with a higher CPV score will present higher levels of PV and CV than the high CPV group of girls.*


## 2. Materials and Methods

### 2.1. Participants

The participants in the study were 1318 adolescents, of whom 14 were excluded due to omissions in the answers related to the variables analysed in this study. The final sample consisted of 1304 adolescents (53.14% girls), aged between 11 and 18 years (M = 13.88, SD = 1.32), enrolled in Compulsory Secondary Education centres in Aragon, Andalusia and the Valencian community. Of these, 24.7% were enrolled in the first year, 27.3% in the second, 23.7% in the third and 24.3% in the fourth. Probability sampling was carried out to select the sample using as primary sampling units the urban geographic areas of the provinces of Alicante, Valencia, Seville and Teruel and as secondary units the public institutes of each area. The socioeconomic level of the areas and centres was average. As regards the sociocultural level of the families, they had intermediate and higher education.

### 2.2. Instruments

Child-to-parent violence. The Child-to-Parents Aggression Questionnaire (CPAQ) [58], adapted from the original [59], was applied. The scale comprises 20 parallel items, 10 referring to the father and 10 to the mother, three of them measuring physical violence (e.g., hitting, kicking) and another seven psychological violence (e.g., insulting, threatening, taking money without permission). The adolescents indicated how often they had carried out these actions against the father or mother in the last year using a four-point Likert scale: 0 (never), 1 (it has occurred once or twice), 2 (it has occurred between three and five times) and 3 (it has occurred six times or more). Cronbach’s alpha was 0.85 for the entire scale.

School victimization. The School Victimization Scale [14], adapted from the original scale [60], was applied. This scale is composed of 22 items that measure three dimensions: relational victimization (e.g., a classmate has ignored me or treated me with indifference), overt physical victimization (e.g., a classmate has beaten me) and overt verbal victimization (e.g., I have been insulted by a classmate). Cronbach’s alpha was 0.92 (relational victimization), 0.67 (overt physical victimization) and 0.88 (overt verbal victimization). 

Cybervictimization. The Adolescent Victimization through Mobile Phone and Internet Scale (CYBVIC-R) [61] was applied. This scale consists of 24 items that measure mobile phone and Internet victimization in two dimensions: relational cybervictimization (e.g., I have been removed or blocked from groups to leave me friendless) and overt cybervictimization (e.g., I have been threatened with calls or voice messages on my mobile). Cronbach’s alpha reliability coefficient was 0.88 for both overt and relational cybervictimization.

### 2.3. Procedure

The data were collected in the year 2018. Initially, the management of the centres was contacted and, once they confirmed their interest and voluntary participation, the objectives and scope of the research were explained. Then, the consent of the families was requested for their children to participate in the study. Afterwards, the data were collected during one 55-minute session held in the usual classroom. Participants were informed of the voluntary nature of their participation and of the possibility of being able to abandon the study at any time and of the guaranteed confidentiality and anonymity of their responses. Two people from the research team remained in the classroom to ensure that the questionnaires were completed properly and to resolve any doubts. This research was carried out taking into account the fundamental principles included in the Declaration of Helsinki in accordance with the ethical values required in research with human beings, as well as subsequent updates [62]. The study was approved by the CEI Ethical Committee of the Virgen Macarena and Virgen del Rocío University Hospitals (CEI VM-VR_01 / 2021_N).

### 2.4. Analysis of Data

The data were analysed with the statistical program SPSS Statistics (v20, IMB, Armonk, NY, USA). A multivariate analysis of variance (MANOVA, 3 × 2) was performed. The independent variables were: CPV with three conditions—high (scores equal to or greater than the 75th percentile), medium (scores below the 75th percentile and greater than 25th percentile) and low (scores less than or equal to the 25th percentile); and sex, i.e., boys and girls. The following dependent variables were selected: relational PV, overt physical PV and overt verbal PV, as well as two conditions of CV; relational and overt. Then, ANOVA was performed to analyse the statistical significance of the variables and the Bonferroni post hoc test was applied (α = 0.05).

## 3. Results

### 3.1. Main Effects

Three groups of adolescents were identified: low CPV (n = 200, 15.3%), moderate CPV (n = 743, 57%) and high CPV (n = 361, 27.7%). Table 1 shows the distribution of CPV in adolescents (low, moderate and high) according to sex (boy or girl).

The MANOVA revealed statistically significant differences in the main effects of CPV (Λ = 0.946, F (10, 2588) = 7317, *p* < 0.001, η_p_^2^ = 0.027) and sex (Λ = 0.892, F (5, 1294) = 31,277, *p* < 0.001, η_p_^2^ = 0.108). A statistically significant interaction effect was observed between sex and CPV (Λ = 0.985, F (10, 2588) = 1915, *p* = 0.039, η_p_^2^ = 0.007).

As regards CPV, the ANOVA revealed significant differences in relational CV (F (2, 1298) = 25.97, *p* < 0.001, η_p_^2^ = 0.038), overt CV (F (2, 1298) = 16.94, *p* < 0.001, η_p_^2^ = 0.025), relational PV (F (2, 1298) = 7.96, *p* < 0.001, η_p_^2^ = 0.012), overt physical PV (F (2, 1298) = 3.13, *p* < 0.05, η_p_^2^ = 0.005) and overt verbal PV (F (2, 1298) = 14.35, *p* < 0.001, η_p_^2^ = 0.022).

Table 2 shows significant differences between the three CPV groups in relational and overt CV, with the high CPV group presenting the highest levels of both types of CV compared to the other two CPV groups. Adolescents with moderate levels of CPV were cybervictimized, in both a relational and overt way, to a greater extent than adolescents with low scores in CPV. Significant differences were also observed in the three groups of CPV in relational, overt physical and overt verbal PV; the high CPV group presented the highest levels of the three types of PV, followed by the group of moderate CPV and then the low CPV group.

As regards sex, the ANOVA showed significantly higher scores than boys in overt CV (F (1, 1298) = 9.29, *p* < 0.01, η_p_^2^ = 0.007) and in girls in relational PV (F (1, 1298) = 16.78, *p* < 0.001, η_p_^2^ = 0.013). On the other hand, significant differences were observed in overt physical PV (F (1, 1298) = 24.71, *p* < 0.001, η_p_^2^ = 0.019), this being higher in the case of boys, as shown in Table 3.

### 3.2. Interaction Effects

A statistically significant interaction effect was observed between CPV and sex in the overt CV variable (F (2, 1298) = 5.56, *p* < 0.01, η_p_^2^ = 0.008). As can be seen in Table 4 and in Figure 1, the results of the post hoc contrast performed with the Bonferroni test (α = 0.05) indicated that the boys who presented high CPV obtained the highest scores in overt CV compared to the other groups, between which no significant differences were found.

## 4. Discussion

The aim of the present study was to analyse the relationships between PV, CV and CPV, taking into account differences according to sex.

The results obtained for the main effects confirmed H1 of this study. Adolescents who scored high on CPV reported higher levels of relational and overt PV, both physical and verbal, as well as higher levels of CV (relational and overt) than adolescents with moderate levels of CPV. Furthermore, this moderate CPV group obtained higher scores in PV and CV than the low CPV group. These results match previous studies reporting that suffering victimization at school was a risk factor for developing violent behaviour at home [63] and exercising CPV [7,55]. A positive school climate has also been linked to less participation in any bullying role [64]. However, this is contradicted by studies that conclude that perceived conflict in the family involved submissive and internalizing behaviours rather than externalizing or aggressive ones [44,45,52,53].

Teens exercising CPV show emotional lability and have difficulty identifying, regulating and expressing emotions [51,65,66], fundamental aspects of emotional intelligence. In this sense, various authors have observed that the management of emotions helps to solve problems and facilitates adaptation to the environment [67,68]. Therefore, these adolescents will have more difficulties relating to their peers and avoiding victimization.

In addition, the lack of resources and strategies to adaptively deal with situations of CV leads adolescents suffering such violence to experience negative feelings, such as frustration, anger, fear and rage [25,69]. These negative emotions can induce victims to release the tension they feel through violence, thus becoming aggressors [6]. This would imply a two-way process; CPV could be the cause of VE and CV but also a consequence.

According to the hypothesis of displaced aggression [70], when a provocation is experienced that excludes the possibility of retaliation, aggressiveness is shifted towards an innocent person or object other than the person responsible for the initial provocation. The person being provoked fears possible retaliation for confronting the initial source and, for this reason, takes his or her frustration out on a person who does not provoke so much fear.

PV and CV experienced by adolescents may be the initial provocation, causing them discomfort. This, coupled with inadequate conflict resolution models, the scarcity of coping resources and the low self-control presented by these victims [24], can favour the displacement of aggression towards parents. This is related to the theory of frustration-aggression [70]. Thus, when faced with PV and CV, adolescents can compensate for the frustration and helplessness they experience by transferring it to the family system and transforming it into violence towards parents [71,72,73,74].

Regarding differences according to sex, boys obtained higher scores than girls in overt physical PV. The fact that boys are more involved than girls in behaviours of overt physical violence has been confirmed by numerous studies [75,76,77]. Therefore, it is consistent that boys are also more victimized in an overt manner than girls.

Boys also obtained higher scores in overt CV; our results indicate that boys overtly victimised in school also suffered overt CV in cyber space. However, most of the studies consulted conclude that CV is more frequent in girls than in boys [78,79,80]. The lack of consensus is probably due to the fact that none of the studies cited differentiated between overt and relational CV. Therefore, this result opens up a new study path to take into account for future research.

The girls suffered more relational PV than the boys, which is congruent with studies that affirm that they are less likely to suffer overt PV, both physical and verbal [26,81,82]. However, it is also worthwhile considering the possibility that perhaps it is not that girls suffer more relational PV but rather that they have a greater capacity than boys to perceive this type of victimization [83].

In contrast with the results obtained for PV, no differences were observed between boys and girls regarding relational CV. The causes of the development of this type of victimization are probably less related to the victim’s sex and more with the cybernetic context itself, which facilitates CV; the aggressor remains anonymous, thus liberating the aggressor from social pressure to behave according to their sex. In addition, there is a physical and emotional distance from the victim, making it more difficult to empathize with the latter.

In terms of the interaction effects considered in H2, a statistically significant effect was observed between CPV and sex in the overt CV variable, where boys with high CPV scores suffered overt CV to a greater extent than other groups. No differences were found between the other groups studied. These results reinforce the previous statement regarding boys, who suffer more overt CV, with the added particularity that this is more likely to occur when adolescents also exercise high CPV.

In CV, the victim does not always know the aggressor, since the latter tends to remain anonymous and, whenever the source of provocation of the discomfort is not available or is an intangible source, the displacement of aggression is aggravated [70,84], in this case towards the parents.

Moreover, boys are more susceptible than girls to public rejection and fear more the consequences of not being integrated into a group [85,86]. As a result, manifest CV causes them major discomfort. For all these reasons, boys who experience manifest CV feel more need than any other group to release their anger and frustration and, since they do not know the identity of the person who initially causes their discomfort, they displace this towards their parents.

Adolescents learn the social skills they need to interact with others in the family context and that is where social learning takes place, through observation and imitation of meaningful models [87]. Good family functioning promotes the adjustment of adolescents, since they learn to develop strategies to interact adaptively with others and this protects them from suffering PV and CV [87,88,89,90]. However, the fact that adolescents perceive conflict in the family makes it easier for them to come across as vulnerable to their peers and be victimized [26]. These adolescents who engage in high CPV are used to and normalize violence, which means they may get involved in virtual contexts in which there are conflictive interactions where it is easier to be the object of aggression.

It would be interesting to examine the results obtained in greater depth and determine whether, in addition to the fact that boys who exercise high levels of CPV also suffer more CV, they also engage in such victimization, since in this case it would be a process of violence-victimization with cyber-victims who are, in turn, cyber-aggressors.

This study has several limitations. All the data collected came from a single source: the adolescents surveyed. It would be worthwhile complementing this information with data from other key informants, such as family members and teachers. It would also be interesting to take into account the age difference of adolescents for future research. This study is based on a cross-sectional design and neither the direction of the relationships nor the causality between the variables analysed could be established. It would be interesting to propose longitudinal studies in the future to identify the victimization journeys followed by adolescents who suffer PV and CV during this stage of the life cycle and their relationship with CPV.

## 5. Conclusions

The results of this research represent important progress necessary to address a problem that causes great social concern. These conclusions highlight the importance of strengthening the psychological adjustment of adolescents, improving relationships with family and with the peer group, as well as taking into account gender socialization processes. In our opinion, intervention programs must be implemented to provide the tools and skills necessary to avoid violence in adolescence. This may be accomplished through the collaboration of professionals from different fields.

## Figures and Tables

**Figure 1 ijerph-18-09360-f001:**
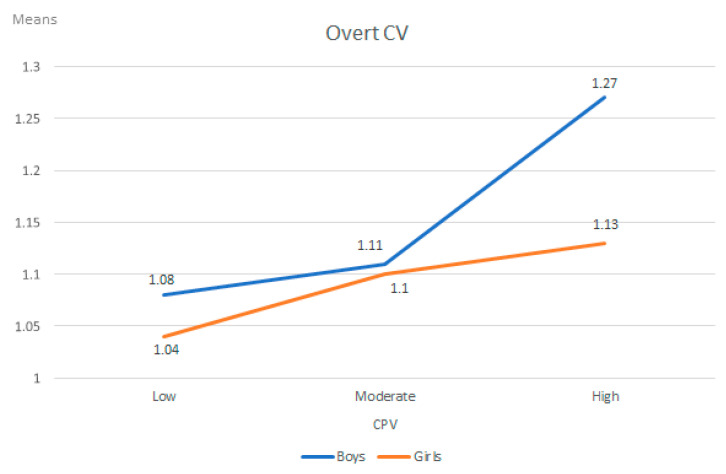
Interaction effect between child-to-parent violence (CPV), sex and overt cybervictimization (CV).

**Table 1 ijerph-18-09360-t001:** Child-to-parent violence (CPV) and sex.

CPV			Low	Moderate	High
Sex	Boys	n	103	363	145
		%	7.9	27.8	11.1
	Girls	n	97	380	216
		%	7.4	29.1	15.6
Total		n	200	743	361
		%	15.3	57	27.7

**Table 2 ijerph-18-09360-t002:** Means, standard deviations and ANOVA results between child-to-parent violence (CPV), cybervictimization (CV) and peer victimization (PV).

		CPV			F (2, 1298)	η_p_^2^
		Low	Moderate	High		
CV	Relational	1.19_c_ (0.44)	1.32_b_ (0.48)	1.51_a_ (0.64)	25.97 **	0.038
	Overt	1.01_c_ (0.23)	1.10_b_ (0.24)	1.18_a_ (0.44)	16.94 **	0.025
PV	Relational	1.57_c_ (0.56)	1.63_b_ (0.57)	1.76_a_ (0.57)	7.96 **	0.012
	Overt physical	1.24_c_ (0.37)	1.26_b_ (0.35)	1.29_a_(0.42)	3.13 *	0.005
	Overt verbal	1.60_c_ (0.61)	1.73_b_ (0.58)	1.87_a_ (0.61)	14.35 **	0.022

Note: Mean (SD); * *p* < 0.05; ** *p* < 0.001; a > b > c.

**Table 3 ijerph-18-09360-t003:** Means, standard deviations and ANOVA results between sex, cybervictimization (CV) and peer victimization (PV).

		Sex			η_p_^2^
		Boys	Girls	F (1, 1298)	
CV	Relational	1.33 (0.50)	1.38 (0.57)	0.64 ^ns^	0.000
	Overt	1.14 (0.35)	1.10 (0.26)	9.29 *	0.007
PV	Relational	1.58 (0.52)	1.72 (0.61)	16.78 **	0.013
	Overt physical	1.32 (0.41)	1.21 (0.33)	24.71 **	0.019
	Overt verbal	1.77 (0.58)	1.73 (0.62)	0.91 ^ns^	0.001

Note: Mean (SD); * *p* < 0.01; ** *p* < 0.001; ns = non-significative.

**Table 4 ijerph-18-09360-t004:** Means, standard deviations and ANOVA results between child-to-parent violence (CPV), sex and overt cybervictimization (CV).

	Sex	CPV			F	η_p_^2^
		Low	Moderate	High		
Overt CV	Boys	1.08 (0.30)_b_	1.11 (0.26)_b_	1.27 (0.53)_a_	F (2, 1298) = 5.56 *	0.008
	Girls	1.04 (0.09)_b_	1.10 (0.22)_b_	1.13 (0.36)_b_		

Note: * *p* < 0.01; a > b.

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
