# Peer review of "Child-to-Parent Violence, Peer Victimization and Cybervictimization in Spanish Adolescents"

_ijerph, 2021, doi:10.3390/ijerph18179360_

Round 1

Reviewer 1 Report

The manuscript explores an important issue, however I think need do explore some issues before it is ready for publication. I add some information in the attached file.

The authors used sex/ gender in an unclear way.

I also recommend the authors to clarify the moment when the data was collected, if before or during the pandemic situation. I suggest the author to reflect about this concern.

Author Response

Dear Reviewer:

Thank you very much for your thorough review and clarity in pointing out errors. We fully agree with the points made and they have already been corrected.

Point 1: I also recommend the authors to clarify the moment when the data was collected, if before or during the pandemic situation. I suggest the author to reflect about this concern

Response 1: The data was collected in the year 2018, before the COVID-19 pandemic. Due to all the inconveniences caused by this situation, the submission of the article for publication was delayed.

Point 2: How where the participants distributed regarding age?

Response 2: When we collected the data, the age of the participants was taken into account, with the idea of grouping them according to three intervals; early, middle and late adolescence. Later, it was decided to simplify the study and not consider the differences based on age; however, we will take it into account for future research. We have added this to the limitations section of the article.

Point 3: Was the research submit to a ethical committee?

Response 3: The study was approved by the CEI Ethical Committee of the Virgen Macarena and Virgen del Rocío University Hospitals (CEI VM-VR_01 / 2021_N)

Point 4: p=.39 stadistically-sigfinicant

Response 4: When in the results section we affirmed that p= .39 was not statistically-significant, it was indeed a mistake in the text that occurred in the translation; p it is stadistically-significant.

We hope that the corrections seem appropriate. We would like to thank the reviewer for his/her comments, which have enabled us to improve our manuscript for publication in IJERPH

Thank you very much, cordial greetings.

Paula López-Martínez

Reviewer 2 Report

Thank you for the opportunity to review your paper entitled “Child-to-parent violence, peer victimization and cybervictimization in Spanish adolescents”

Abstract and Material and Methods

Indicate the type of study.

Material and Methods

How was the sample size calculated?

What was the response rate?

Reviewer 3 Report

The aim of this work was to examine the relationship between child-to-parent violence (CPV), peer and cybervictimization (PV, CV) in adolescents taking into account gender. 

I think that the article suppose a good contribution to scientific community. However, there are some issues that need to be addressed to better justifiy the aims and hypotheses and to improve the discussion of the results.  

The authors analyse the relationship between one way of aggression (CPV) and two ways of victimization (PV and CV). However in the introduction section, it seems that they justify the relationship between all the variables as they were analysing the relationship between three ways of victimization. In line 105-107 can be read "However, the perception of family conflict contributes to the fact that adolescents are victims of victimization, since they tend to perform submissive behaviours and present themselves to their peers as vulnerable and easy targets for abuse". Authors need to justify better why hypothesize that CPV is related to CV and PV trying to explaing the way of relationship between these variables. This is important because in the discussion, sometimes they explain CPV as a possible risk factor/cause of CV and PV (line 334-336) and also as a consequence (line 278-283). Even although the aim of the study was not to show an explanative model and the design of the study (transversal) hinders the establishment of causal relationship, they need to explain as clear as possible what previous studies suggest regarding the relationshipt between all the constructs to be able to discuss the results in a appropriate way and to justify hypotheses. 

Furthermore, regarding the family profile of victims of bullying or cyberbullying (characterized by the perception of parenting practices such as high levels of psychological control, low promotion of autonomy and the use of punitive discipline with some gender differences) is hard, although not ilogical, to think that these same students can become in aggressor of their parents when, in fact, evidence suggest that they have perceived some type of abussive behaviors from them, being the more common submissive and internalizing reactions more than agressive or externalizing behaviors against them (Gómez-Ortiz et al., 2016; 2018, 2019; Nocentini et al., 2019). In this sense, studies that indicate this trend have also to be taken into account in the discussion and not only those which confirm the hypotesis.  

I think that there are two mistakes: VFP line 266 and VE line 288. 

References:

  • Gómez-Ortiz, O., Romera-Félix, E. M., & Ortega-Ruiz, R. (2016). Parenting styles and bullying. The mediating role of parental psychological aggression and physical punishment. Child Abuse & Neglect, 51, 132-143. http://dx.doi.org/10.1016/j.chiabu.2015.10.025 Ver post-print de autor 
  • Gómez-Ortiz, O., Romera-Félix, E. M. Ortega-Ruiz, R., & Del Rey, R. (2018). Parenting practices as risk or preventive factors for adolescent involvement in cyberbullying: Contribution of children and parent gender. International Journal of Environmental Research and Public Health15. https://doi.org/10.3390/ijerph15122664
  • Gómez-Ortiz, O., Apolinario, C., Romera-Félix, E. M. & Ortega-Ruiz, R. (2019). The role of family in bullying and cyberbullying involvement: Examining a new typology of parental education management based on adolescents’ view of their parents. Social Sciences, 8. https://doi.org/10.3390/socsci8010025
  • Nocentini, A., Fiorentini, G., Di Paola, L., Menesini, E. (2019). Parents, family characteristics and bullying behavior: A systematic
    review. Agression and Violent Behavior, 45, 41-50. https://doi.org/10.1016/j.avb.2018.07.010

Author Response

Dear Reviewer:

Thank you very much for your thorough review and clarity in pointing out errors. We fully agree with the points made and they have already been corrected. We also appreciate the references of studies that conclude results different from ours, which we have incorporated both in the introduction and in the discussion of the article.

Point 1: Authors need to justify better why hypothesize that CPV is related to CV and PV trying to explaing the way of relationship between these variables. In the discussion, sometimes they explain CPV as a possible risk factor/cause of CV and PV.

Response 1: First, both adolescents who exercised CPV, as well as those who suffered PV and CV reported family conflicts, such as lack of communication and little emotional support. Some studies had already linked family conflicts in general and, in particular, CPV, with the fact of suffering victimization. This is because one of the consequences of experiencing conflict in the family is that adolescents lack the necessary resources to cope with victimization and also tend to react passively, which makes them easy targets for abuse. However, in families where conflict is not perceived, there is good affection and communication, adolescents have effective strategies to deal with victimization, which they suffered to a lesser extent. Few studies had studied the link between CPV and CV and, above all, we were interested in knowing how suffering CPV affected the different types of PV and CV. Therefore, one of our starting hypotheses was "Adolescents with high levels of CPV will suffer more PV, both relational and overt, physical and verbal, as well as more CV, both relational and overt". However, already in the bibliographic review we found the possibility, which we later discussed, that probably, the relationship between CPV and victimization is a two-way process; CPV is both a cause and a consequence of victimization. This may be because the negative emotions experienced by adolescents suffering from PV and CV imply a frustration that they do not feel capable of venting with their peers, but with their parents.

We are aware that the justification was not properly clarified in the initial manuscript, we have made the pertinent modifications both in the introduction and in the discussion and we hope that now it is clearer what we initially intended to express.

Point 2: Regarding the family profile of victims of bullying or cyberbullying (characterized by the perception of parenting practices such as high levels of psychological control, low promotion of autonomy and the use of punitive discipline with some gender differences) is hard, although not ilogical, to think that these same students can become in aggressor of their parents when, in fact, evidence suggest that they have perceived some type of abussive behaviors from them, being the more common submissive and internalizing reactions more than agressive or externalizing behaviors against them. In this sense, studies that indicate this trend have also to be taken into account in the discussion and not only those which confirm the hypotesis.

Response 2: It is true that the family profile of victims and cyber victims implies internalizing and submissive behaviors, but there are other adolescents who react in an aggressive and externalizing way. A percentage of the victims of bullying and cyberbullying do not react passively, but rather reactively, attacking their peers; They are called "bully-victims", who have an even worse psychological prognosis than adolescents who are only victims or only aggressors.

It is true that we had not taken into account the studies that associated victimization with these internalizing and submissive behaviors, so we have incorporated them into the manuscript and discussed it.

We hope that the corrections seem appropriate. We would like to thank the reviewer for his/her comments, which have enabled us to improve our manuscript for publication in IJERPH.

Thank you very much, cordial greetings.

Paula López-Martínez

Reviewer 4 Report

It is a well thought out and well developed study. The method is rigorous and the results are adequately discussed within the scope of the research questions posed in the introduction to the study. The bibliographic references are excessive (more than 100) for a research of this type. It is recommended to reduce them, especially with regard to conference citations and others less accessible to the scientific community.

Author Response

Dear Reviewer:

Thank you very much for reading our article and for the suggestions made.

Point 1: The bibliographic references are excessive (more than 100) for a research of this type. It is recommended to reduce them, especially with regard to conference citations and others less accessible to the scientific community.

Response 1: We have taken their suggestion into account and, despite having introduced new references at the suggestion of other reviewers, we have reduced the total number of them. We have also removed the conference citation.

We hope that the corrections seem appropriate. We would like to thank the reviewer for his/her comments, which have enabled us to improve our manuscript for publication in IJERPH.

Thank you very much, cordial greetings.

Round 2

Reviewer 3 Report

All is ok. 

Author Response

Dear reviewer,

Thank you very much for your thorough review.

Kind regards.